

# Accurate Assessment of Land-Atmosphere Coupling in Climate Models Requires High Frequency Data Output

Kirsten L. Findell[1], Zun Yin[1,2], Eunkyo Seo[3,4], Paul A. Dirmeyer[4], Nathan P. Arnold[5], Nathaniel Chaney[6], Megan D. Fowler[7], Meng Huang[8], David M. Lawrence[7], Po-Lun Ma[8], and Joseph A. Santanello Jr.[9]

[1]Geophysical Fluid Dynamics Laboratory, National Oceanic and Atmospheric Administration, Princeton, NJ, 08540, USA
[2]Atmospheric and Oceanic Sciences, Princeton University, Princeton, NJ, 18966, USA
[3]Department of Environmental Atmospheric Sciences, Pukyong National University, Busan, 48513, Republic of Korea
[4]Center for Ocean-Land-Atmosphere Studies, George Mason University, Fairfax, VA, 22030, USA
[5]NASA-GSFC, Global Modeling and Assimilation Office, Greenbelt, MD, 20771, USA
[6]Department of Civil and Environmental Engineering, Duke University, Durham, NC, USA
[7]Climate and Global Dynamics Laboratory, National Center for Atmospheric Research, Boulder, CO, 80305, USA
[8]Atmospheric Sciences and Global Change Division, Pacific Northwest National Laboratory, Richland, WA 99354, USA
[9]NASA-GSFC, Hydrological Sciences Laboratory, Greenbelt, MD, 20771, USA

*Correspondence to*: Kirsten L. Findell (Kirsten.Findell@noaa.gov)

**Abstract.** Land-atmosphere (L-A) interactions are important for understanding convective processes, climate feedbacks, the development and perpetuation of droughts, heatwaves, pluvials, and other land-centred climate anomalies. Local L-A coupling
(LoCo) metrics capture relevant L-A processes, highlighting the impact of soil and vegetation states on surface flux partitioning, and the impact of surface fluxes on boundary layer (BL) growth, development, and entrainment of air above the BL. A primary goal of the Climate Process Team on Coupling Land and Atmospheric Subgrid Parameterizations (CLASP) is parameterizing and characterizing the impact of subgrid heterogeneity in global and regional earth system models (ESMs) to improve the connection between land and atmospheric states and processes. A critical step in achieving that aim is the
incorporation of L-A metrics, especially LoCo metrics, into climate model diagnostic process streams. However, because land-atmosphere interactions span time scales of minutes (e.g., turbulent fluxes), hours (e.g., BL growth and decay), days (e.g., soil moisture memory), and seasons (e.g., variability of behavioural regimes between soil moisture and latent heat flux), with multiple processes of interest happening in different geographic regions at different times of year, there is not a single metric that captures all the modes, means, and methods of interaction between the land and the atmosphere. And while monthly means
of most of the LoCo-relevant variables are routinely saved from ESM simulations, data storage constraints typically preclude routine archival of the hourly data that would enable the calculation of all LoCo metrics.
Here we outline a reasonable data request that would allow for adequate characterization of sub-daily coupling processes between the land and the atmosphere, preserving enough sub-daily output to describe, analyse, and better understand L-A coupling in modern climate models. A secondary request involves embedding calculations within the models to determine





mean properties in and above the BL to further improve characterization of model behaviour. Higher-frequency model output

will (i) allow for more direct comparison with observational field campaigns on process-relevant time scales, (ii) enable

demonstration of inter-model spread in L-A coupling processes, and (iii) aid in targeted identification of sources of deficiencies

and opportunities for improvement of the models.

**1 Introduction**

Much progress has been made in understanding and characterizing land-atmosphere (L-A) interactions in recent years (for an

overview of some advances, see Santanello et al., 2018). The importance of L-A interactions has been demonstrated in the

initiation, perpetuation, propagation and termination of droughts (e.g., Otkin et al., 2018; Roundy et al., 2013; Herrara-Estrada

et al., 2019; Wu and Dirmeyer 2020), in the exacerbation of heat waves (Findell et al., 2017; Alizadeh et al. 2020; Petch et al.

2020; Selten et al. 2020; Seo et al. 2020; Dirmeyer et al., 2021; Benson and Dirmeyer 2021), and in the timing of monsoon or

rainy season onset (e.g., West Africa: Berg et al., 2017; India: Tuinenberg et al. 2014; Amazon: Wright et al., 2017). These

and other studies collectively suggest the importance of accurately modelling processes at the heart of these feedbacks and

interactions. However, output from climate model simulations is rarely saved at high enough frequencies to capture the rapidly

changing features and fluxes that are crucial to the proper characterization of the many links in the chain of L-A interactions

(Santanello et al., 2011). These individual linkages include:

• The impact of surface temperature, soil moisture and vegetation on turbulent fluxes at the L-A interface,

   • The impact of those fluxes on boundary layer (BL) mixing and moist static energy (MSE),

   • The impact of BL processes (e.g., growth rate and buoyancy) on entrainment of air above the BL, and

   • Their cumulative impact on

     o the BL height, temperature, and humidity, and

o the development of clouds and/or precipitation.

Figure 1 schematically demonstrates that individual metrics of L-A coupling capture different aspects of these complex

linkages. While some metrics focus on the physical processes that operate within the diurnal cycle (e.g., mixing diagrams,

Santanello et al., 2009, 2011), others focus on the signal of L-A interactions emerging from long-term multi-variate statistics

(e.g., the triggering feedback strength or TFS, Findell et al., 2011). Because of this complexity, we cannot select just one

variable, metric, or timescale to assess the strength of a model's coupling between the land and the atmosphere.

The objects in Figure 1 highlight the distinction between metrics that elucidate physical processes directly (within the diurnal

cycle) and those that look at the statistical behaviour in data aggregated into long time series, using sub-daily, daily, or longer-

term mean values in the statistical analyses. Both classes of metrics provide useful information about L-A coupling; when used

to inform model development and improvement, the statistical metrics can reveal symptoms of model behaviour, while the

process-oriented metrics can potentially diagnose causes. (See Neelin et al., (2023) for a detailed appreciation and application

of process-oriented diagnostics to assess and improve model behaviour). For the purposes of demonstrating some of the critical





information that can be learned from analysing observations and models at sub-daily time scales, we will focus on the use of mixing diagrams (Santanello et al., 2009, 2011), two-legged metrics at multiple scales (Dirmeyer et al., 2011; Yin et al., 2023; Seo and Dirmeyer, 2022), and the Triggering Feedback Strength (TFS, Findell et al., 2011, 2015).

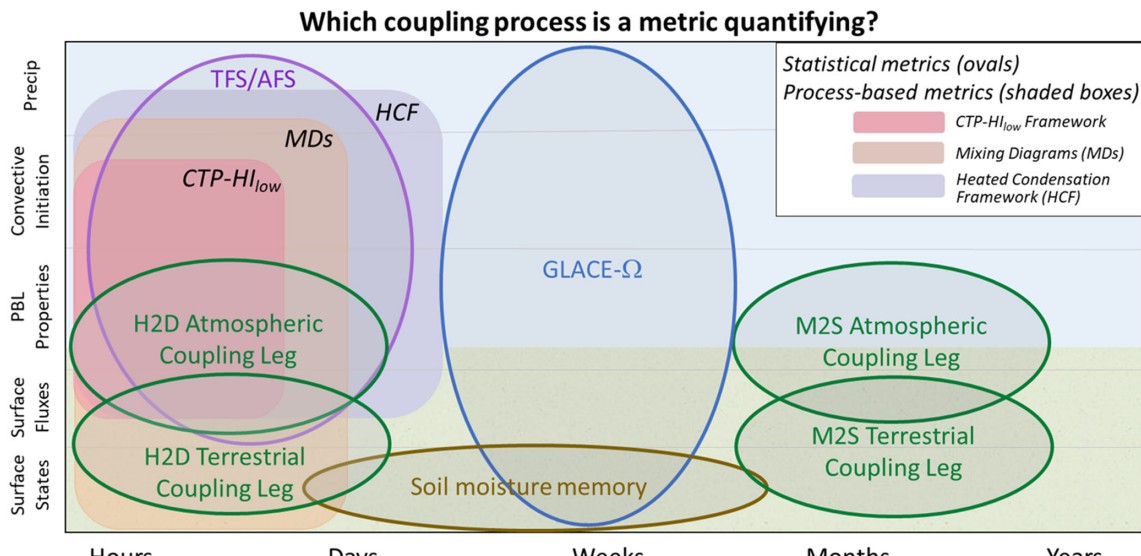


**Figure 1: LoCo metrics assess interactions between different parts of the earth system (y-axis) at different temporal (x-axis) scales. Yin et al. (2023) and Seo and Dirmeyer (2022) highlight the need to recognize that the two-legged metrics of Dirmeyer et al. (2012) yield results that are dependent on the temporal frequency of the input data, thus requiring a separation between hourly-to-daily (H2D) and monthly-to-seasonal (M2S) versions of the two-legged metrics. (Modified from Santanello et al., 2018.)**


Recent observational field campaigns have included high-frequency observations that can be compared to output from models covering a wide range of purposes and scales (e.g., ESMs, regional climate models, large eddy simulations, and single-column land-atmosphere models) to test assumptions about L-A behaviour. These include the Land–Atmosphere Feedback Experiment (LAFE) at the Southern Great Plains (SGP) site near Lamont, Oklahoma, USA (Wulfmeyer et al., 2018), the Chequamegon

Heterogeneous Ecosystem Energy-balance Study Enabled by a High-density Extensive Array of Detectors (CHEESEHEAD)



in Wisconsin, USA (Butterworth et al., 2021), and the Land surface Interactions with the Atmosphere over the Iberian Semi-arid Environment (LIAISE) experiment in northeastern Spain (Boone et al., 2021). For example, using high-frequency data from three observational towers from LAFE, Wulfmeyer et al. (2022) demonstrate some of the shortcomings of Monin-Obukov similarity theory (MOST, Monin and Obukhov, 1954) in the estimation of surface fluxes of sensible heat, latent heat and
momentum in unstable conditions. The widespread use of MOST in many model parameterizations speaks to the progress enabled by its implementation. However, the recent acquisition of high-frequency observations like those from LAFE and longer-lifespan Land-Atmosphere Feedback Observatories (LAFOs) with the same instrumentation (Späth et al, 2023) expose model shortcoming which can only be evaluated with high-frequency model output. While "high-frequency" in the context of GCMs means something different than in the context of boundary layer turbulence (typically on the order of seconds), the data
request presented here will enable evaluation of processes occurring on hourly to three-hourly time scales, enabling a leap forward in understanding both the processes themselves and ESM representations of those L-A coupling processes.

While short-term simulations saving high-frequency output would allow for comparison of models with field campaigns, to accurately capture the long-term signal of L-A coupling characterized by the statistically-based L-A metrics shown in Figure 1, sub-daily output of fields at the L-A interface must be saved as part of the routine diagnostic output from long simulations.
Furthermore, previous studies have demonstrated that metrics assessing interactions *between* directly observed variables (e.g., TFS is not directly observed, but assesses the relationship between observed fluxes and precipitation) require longer datasets than directly observed variables (e.g., precipitation) to adequately sample the joint parameter space and compute a statistically robust climatology (Findell et al., 2015).

To assess the coupling strength and details of the interactions in different parts of the L-A system of a GCM, a comprehensive
data request would include:

- Hourly 3D atmospheric profiles of potential temperature ($\theta$), humidity ($q$), and three-dimensional winds ($u$, $v$, $w$);
- Hourly 3D soil profiles of moisture content (SM) and temperature ($T_{soil}$); and
- Hourly 2D fields of surface pressure, BL height ($h_{PBL}$), precipitation (P), sensible heat flux (H), evapotranspiration (ET) and its component parts; near-surface (2m) temperature, humidity, and winds; net radiation ($R_{net}$) fields
(incoming and outgoing short- and long-wave radiation: $SW_{down}$, $SW_{up}$, $LW_{down}$, $LW_{up}$), and land surface temperature (LST).

The atmospheric profiles should cover the region from the surface to the mid-troposphere in order to capture characteristics of air entrained at the top of the BL. The soil profiles should span from the top of the soil column down through the root zone, at a minimum. These data would allow for calculation of a host of LoCo metrics, including all but one of the metrics displayed
in Figure 1, at the time scales that are most relevant to the daytime processes the metrics are meant to describe. (The GLACE-$\Omega$ metric can only be determined with specific model simulations; see Koster et al., 2004.) However, we recognize that this would require copious amounts of archive capacity. Here we aim to reduce that request substantially and include only two-dimensional fields. *Our goal is to define a data request which is reasonable in its storage requirements, but still provides enough information to characterize the core aspects of the sub-diurnal processes central to L-A interactions.* More specifically,



the goal is to define a small but sufficient number of data samples per day from two-dimensional fields capturing the sub-diurnal evolution and variability of:

- Boundary layer properties (BL height; vertically-averaged or representative mixed-layer heat content, humidity, and advection);
- Fluxes and radiation fields (precipitation, sensible and latent heat fluxes, net radiation or individual components);
- A bulk measure of stability and humidity deficit above the BL, and
- Root-zone and/or surface soil moisture and temperature conditions.

In Section 2 we highlight the complexity of the L-A system, showing the many interaction pathways between individual component parts. In Section 3 we demonstrate why sub-daily data are required, use these results to provide substantive rationale for the minimum data frequency required to adequately characterize the sub-daily processes of interest, and share an example

of the type of behaviour that could be routinely assessed if the requested data were regularly made available for model development and/or evaluation. In Section 4 we put forth our data request proposal, followed by conclusions in Section 5.

## 2 Highlighting the complexity of the land-atmosphere system

The novel "pipe diagrams" in Figure 2 compile linkages as coupling strength indices assessed from daily summertime (June-July-August, JJA) data at the AmeriFlux tower at the SGP field site, along with corresponding diagrams from two versions of

the National Oceanic and Atmospheric Administration (NOAA) United Forecast System (UFS) model for the grid cell closest to the SGP site. These coupling strength indices are modelled after the two-legged metrics, named in recognition of the two phases of interaction: the terrestrial leg, which assesses the connection between soil moisture and surface fluxes, and the atmospheric leg, which focuses on the connection between surface fluxes and the BL (Figure 1; Dirmeyer et al., 2011, 2014). Pipe diagrams from approximately 170 flux tower locations were used during recent model development, aiding the evaluation

of UFS Prototype 6 (P6) to Prototype 7 (P7) (Seo et al., 2023). An advantage of these diagrams is the ability to visualise a host of different L-A linkages at once, and thus identify systematic model biases or behaviours.

The individual coupling strength indices in Figure 2 are all indicative of both the sensitivity of a target variable, $T$, (e.g., latent heat flux), to a source variable, $S$, (e.g., soil moisture), and the amount of observed variability:

$$\sigma(T)r(S,T), \tag{1}$$

where $\sigma(T)$ is the daily standard deviation of the target variable, and $r(S,T)$ is the correlation between the two variables. In each pipe diagram, the absolute value of this index is proportional to the width of the link; the strength of the $\sigma(T)$ term is listed under each variable name, and is visually revealed through the intensity of the green colour in the box around the variable name; the strength of the correlation term is enumerated with "$r=$" on each link, and is visually represented by the colour intensity of the link. The physically-expected sign of the correlation between each source and target variable is given by a red

triangle on the link when a positive correlation is expected (e.g., high soil moisture is associated with high latent heat flux), and by a blue triangle when a negative correlation is expected (e.g., high sensible heat flux is associated with low evaporative fraction: $EF = \lambda E / (H + \lambda E)$, i.e., where $H$ is sensible heat flux and $\lambda E$ is latent heat flux, with $E$ being the evaporation rate





and $\lambda$ the latent heat of vaporization). When the calculated correlation is of the opposite sign from this expectation, then the variability of the source term is not driving the variability of the target term, so the feedback is "severed" and the link is

represented with a dashed line.

Comparing Figures 2a and 2b quickly reveals that at the grid cell closest to the SGP site, the UFS P6 model exhibits stronger variability in surface fields and stronger coupling between the soils (both moisture and temperature) and the fluxes than is measured at the observational flux tower. Additionally, the modelled fluxes exhibit stronger coupling to 2m humidity and temperature than the observations show. Though observations have inherent uncertainties from measurement error and issues

associated with the representativeness of a single point to the broader region characterized by the model grid cell, this information was used during the model development process, with changes being made to both the land model (Noah in UFS P6 to Noah-MP in UFS P7) and the boundary layer parameterization to improve the full spectrum of coupling strengths manifesting in UFS P7 (details of changes between UFS prototype versions are provided in Stefanova et al., 2022). As a result, the UFS P7 pipe diagram in Figure 2c is a better match to the observations of Figure 2a, than that of UFS P6 in Figure 2b.

Pipe diagrams like Figure 2 can be extended vertically to include additional physical fields and states, accounting for additional links in the LoCo process chain (Santanello et al., 2018). For example, BL properties could include average BL potential temperature, humidity, or moist enthalpy, BL height, and height or pressure of the lifted condensation level (LCL). A final layer at the top of these pipe diagrams could include information about clouds and precipitation. The myriad of possible links in the process chains connecting individual elements within these pipe diagrams, and indeed within the physical land-

atmosphere system, demonstrate the complexity of interactions between the land and the near-surface atmosphere. Figure 2 demonstrates that model parameterizations influence the modelled strength and connectivity of different parts of the L-A system, and that confronting models with process-level observations from different climate regimes can help expose model deficiencies and limitations. For that to be possible, however, model output must be temporally equivalent to the observations in-hand, and it must adequately sample the behaviour of the physical processes of interest. While daily data were successfully

leveraged to improve land-atmosphere coupling in the UFS model, the next section demonstrates some of the processes requiring sub-daily data.





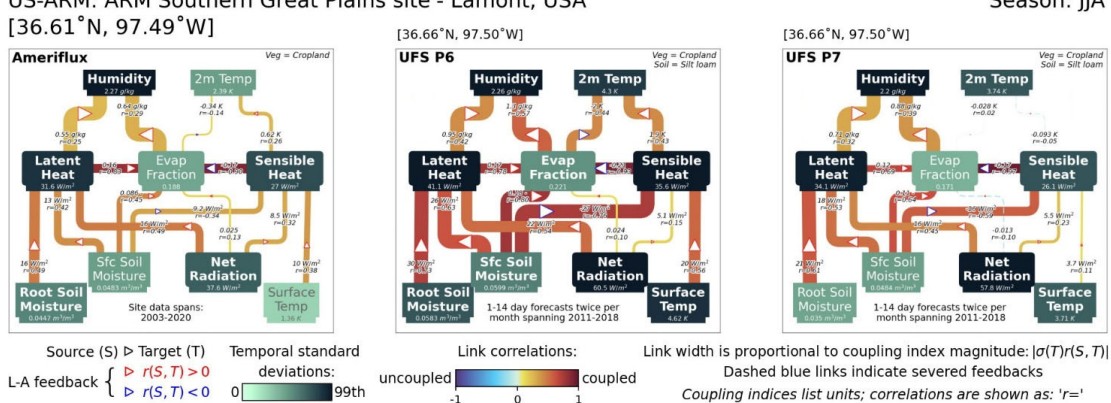

**Figure 2: Land-atmosphere coupling pipe diagrams demonstrating the complexity of land-atmosphere interactions, the need for more than one measure to assess coupling, and some of the potential inadequacies of modelled coupling at this example location.**
**(Taken from http://cola.gmu.edu/dirmeyer/ufs/P6vP7_loco_chains_AMX.html.)**

### 3 Establishing the need for high temporal resolution data

The Triggering Feedback Strength (TFS, Findell et al., 2011) is a measure of the sensitivity of afternoon rainfall occurrence to morning-time evaporative fraction ($EF = \lambda E / (H + \lambda E)$). Using three-hourly data from the North American Regional Reanalysis (NARR; Mesinger et al., 2006), Findell et al. (2011) showed that high morning EF enhances the probability of
afternoon rainfall east of the Mississippi and in Mexico, with higher EF leading to increases in afternoon rainfall probability of between 10 and 25% in these regions. By contrast, the intensity of rainfall was shown to be largely insensitive to surface flux partitioning, as assessed by the Amplification Feedback Strength (AFS; Findell et al., 2011).

A follow-up study by Berg et al. (2013) showed that the Geophysical Fluid Dynamics Laboratory (GFDL) model AM2.1 exhibited similar sensitivity of afternoon rainfall likelihood on morning surface flux partitioning in the eastern US and Mexico,
and a similar insensitivity of rainfall intensity to surface flux partitioning. However, the similar TFS results from AM2.1 and NARR occurred for different reasons. Like the two-legged metrics discussed above (Equation 1), the TFS is computed with a sensitivity term (the sensitivity of the probability of afternoon rain to variations in morning-time EF) multiplied by a standard deviation term ($\sigma_{EF}$). In contrast to the two-legged metrics, however, the calculation of the TFS is a summation of purposefully binned or segmented data to account for the possibility of non-uniform sensitivities in different EF regimes; indeed, sensitivity
strength is substantially larger at EF > 0.6 than at smaller EF values (Findell et al., 2011). Berg et al. (2013) showed that the regions with high TFS values in AM2.1 were driven by larger EF variability (peak $\sigma_{EF}$ values of 0.2 in NARR, compared to 0.4 in AM2.1), while regions with high TFS values in NARR were driven by larger mean rainfall sensitivities (peak mean sensitivities above 2 in NARR, compared to less than 1 in AM2.1). The large values of $\sigma_{EF}$ in the AM2.1 results also explained



an additional region of high TFS values in AM2.1 in the northern central Great Plains of the US, extending into adjacent areas
in southern Canada.

Figure 3 shows the June-July-August TFS (panel a) and its two component parts (panels b and c) calculated from hourly
European Centre for Medium-Range Weather Forecasts 5$^{th}$ reanalysis data (ERA5; Hersbach et al., 2018, 2020). Comparison
with Findell et al. (2011) and Berg et al. (2013) show that NARR, ERA5, and AM2.1 exhibit the same range of sensitivity of
afternoon rainfall triggering to morning-time flux partitioning, but in the ERA5 data, the peak TFS values of 15-25% only
manifest in Mexico with some extension into the southern part of the mountainous US southwest. While the eastern US region
shows up with relatively elevated component contributions in ERA5 (Figures 3b-c), the resultant TFS values are only 5-10%
in most of the eastern US, and approach 15% in much of Florida (Figure 3a). The individual terms contributing to ERA5's
TFS results have peak values matching the smaller EF variability of the NARR data, rather than the high variability of AM2.1
(Figure 3c here and Figure 6 of Berg et al., 2013), and sensitivities matching the smaller AM2.1 values, rather than those of
the NARR data (Figure 3b here and Figure 7 of Berg et al., 2013). These differences across reanalysis datasets are likely
impacted by differences in the data assimilation protocols and observational datasets ingested by ERA5 and NARR. In
addition, the TFS may also be highly sensitive to each system's parameterizations of the surface layer, boundary layer, and
convection, since the surface fluxes at the heart of the TFS are not assimilated variables, but are wholly model dependent
(Kalnay et al., 1996). Additional investigation is necessary to better understand these differences between the reanalyses and
the model, but this behaviour can only be exposed with analysis of sufficiently high-frequency data. Here, data frequencies of
at least 3 hours were essential to enable the separation of morning-time fluxes and afternoon precipitation events.

While a paucity of high-frequency data has forced many previous analyses of two-legged metrics (Equation 1) to rely on
monthly mean data (e.g., Dirmeyer et al., 2014; Hu et al., 2021; Lorenz et al., 2015), Yin et al. (2023) highlight the need to
recognize that the two-legged metrics yield results that are dependent on the temporal frequency of the input data (Figure 4
and the H2D-M2S distinctions in Figure 1), in part because the magnitude of variability is dependent on the averaging period
of the data being analysed, and in part because the inclusion of night time hours can mask the daytime feedbacks that are at
the heart of the sensitivity between the variables of interest. Figure 4 shows that the assessment of the strength of the
atmospheric leg measuring the impact of sensible heat flux, $H$, on BL growth (as assessed by the pressure of the lifting
condensation level, $p_{LCL}$) can be very different when using monthly (M), 24-hour entire-day (E), or daytime-only (D; 0700 to
1500 local time) time series. Different averaging periods of the input data effectively allow one to ask different questions about
coupling: monthly-averaged data tell us about the seasonal variability of the terms being assessed and their coupling, while
daytime-only data are needed to tell us about the direct impact of surface fluxes on BL properties, for example. In regions
where the month-to-month variability is small (e.g., where mean $H$ and $p_{LCL}$ values are similar for all summer months),
substantial day-to-day variability in these terms will not be captured by monthly mean values (e.g., orange regions in Figure
4). However, in regions where the progression into deeper days of summer tends to bring drier and drier conditions, differences
across summer months (e.g., June compared to August) can be substantial, so monthly mean time series will still show high
variability and potentially result in a diagnosis of a large coupling strength (e.g., blue regions in Figure 4). Comparing daily to



sub-daily scales, Figure 4 shows about 30% disagreement in the highlighted regions with strong $H$-$p_{LCL}$ coupling determined from E versus D time series. The night-time component of E was shown to obscure the diurnal coupling signal in some areas,

with complications caused by regionally-specific mechanisms (particularly in the very arid regions adjacent to the Mediterranean Sea) or UTC-based time smoothing (Yin et al., 2023). These differences highlight the need for sub-daily data to accurately capture the process-level connections between surface fluxes and the BL response.

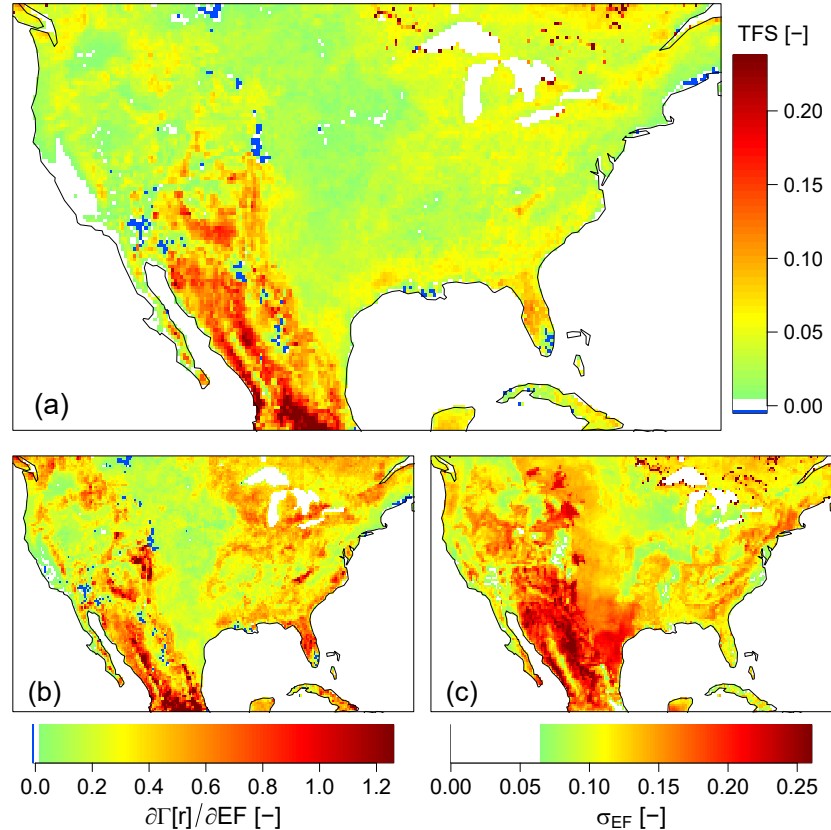

**Figure 3: (a) The Triggering Feedback Strength (TFS; units of probability of afternoon (noon-6 pm) rain) in summer (JJA) based on ERA5 hourly data from 1991 to 2020. The TFS algorithm follows Findell et al. (2015) but with a ten-bin segmentation of daily evaporative fraction (EF). Positive values indicate the morning EF positively affects the probability of the occurrence of afternoon precipitation. (b) The mean value of the sensitivity term contributing to the TFS ($\overline{\partial\Gamma(r)/\partial EF}$), thus the mean sensitivity of afternoon rainfall to morning-time surface flux partitioning. (c) The variability term contributing to the TFS: the standard deviation of EF.**



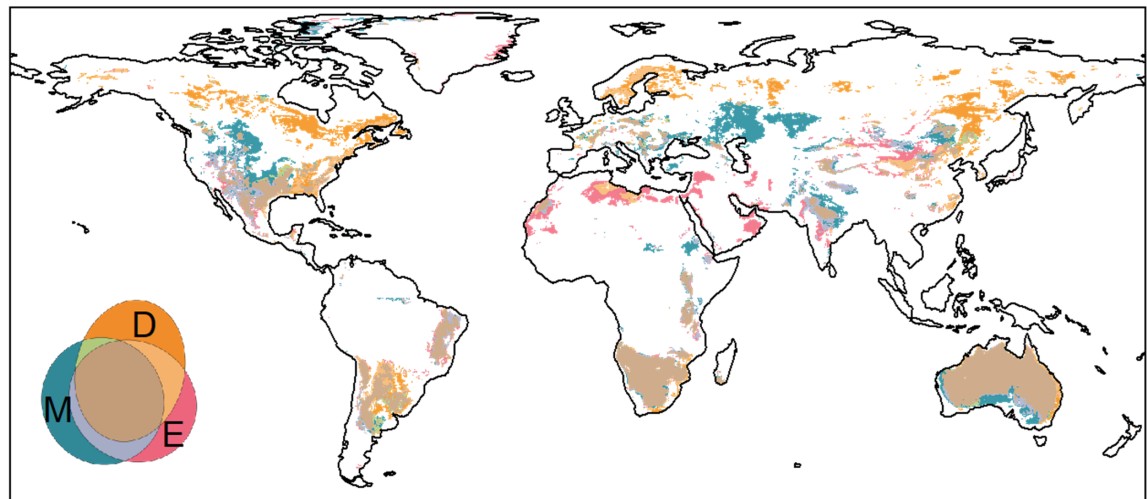


**Figure 4: Two-legged metric analysis demonstrating the impact of different averaging periods on the assessment of coupling strength in summer (JJA and DJF for the North and the South Hemispheres, respectively). The diagnoses are based on the ERA5 (ECMWF ReAnalysis 5) reanalysis data from 1991 to 2020. The coupling strength between the sensible heat flux and $p_{LCL}$ is estimated by the TLM algorithm (Dirmeyer et al., 2006). Strongly coupled regions (top 15% percentile of land grid cells) are diagnosed by using**
**different time series (i.e., D: daytime-only mean; E: 24-hour entire-day mean; and M: monthly mean). The Euler diagram is employed to illustrate the spatial differences between the three diagnoses. The areas of colored components in the Euler diagram are proportional to the sizes of specific sets. (Modified from Yin et al., 2023.)**

Seo and Dirmeyer's (2022) thorough evaluation of the hourly evolution of BL temperature and humidity at flux tower
observational sites can be leveraged to determine the minimum number of data points needed per day to adequately capture both the thermal and the moisture evolution of the BL. Figure 5a shows hourly mixing diagrams spanning all hours of the day, based on Seo and Dirmeyer (2022), plotting moist (x-axis) and heat (y-axis) energy content per unit mass within the mixed layer, averaged across the 10% of the 230 stations that were the most moisture-limited (red circles) and the most energy-limited (blue squares; see Supplemental Figure S1 for a global map with station locations). Through their detailed analysis, Seo and
Dirmeyer highlight differences in the timing of the BL response to moisture fluxes compared to heat fluxes, with the thermal process chain often leading the moist process chain by 2-3 hours during the day, and rapid thermal decoupling in the late afternoon contrasted with a gradual decline of moist coupling throughout the evening hours. They also highlight dependence of the timing of humidity minimums on moisture availability: Figure 5a shows that the driest time for the BL is during early afternoon in moisture-limited regimes, but before sunrise in energy-limited regimes. Both moisture- and energy-limited regions
show a morning time peak in BL humidity (7-9 am).

Findell et al. (2017) showed that some of these behaviours can be captured in a statistical sense using monthly mean diurnal cycles of temperature and moisture, but a full step-by-step understanding of these detailed processes and interactions requires many data points per day. Figure 5b shows that 3-hourly output generally captures the critical phases and the maximum extent



of the diurnal excursions in T-q phase space, as well as the bulk of the diurnal asymmetry of the T-q evolution. The numbers
to the right of each mixing diagram quantify the area within the curve (e.g., 8.26e10[6] J[2]/kg[2] in the water-limited diagram of
Figure 5a, compared to 1.23e10[6] J[2]/kg[2] in the energy-limited composite) and make it clear that while the 3-hourly mixing
diagrams underestimate the diurnal asymmetry, the process-relevant distinction of small asymmetry in energy-limited regimes
compared to large asymmetry in water-limited regimes remains clear. While six-hourly data (Figure 5c) can capture the
approximate timing of the humidity minimums (late afternoon versus early morning), such infrequent sampling can miss the
most rapidly changing portions of the daytime T-q evolution (e.g., samples beginning at 0LST), leading to inaccurate
assessments of the extent of the diurnal asymmetry in T-q energetic phase space.

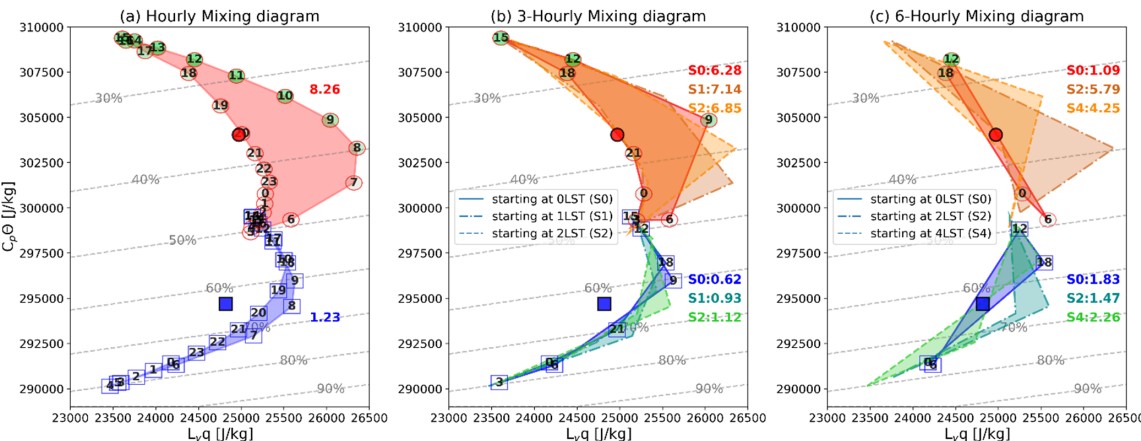

**Figure 5: (a) The hourly mixing diagrams in water- (red) and energy- (blue) limited flux tower sites exhibits the coevolution of**
**moisture (x axis) and thermal (y axis) energy content per unit mass within the BL (Modified from Fig. 5d in Seo and Dirmeyer,**
**2022). The marks are shaded by the color determined by two-legged couplings corresponding to the local hour (referring to Fig. 5a**
**in Seo and Dirmeyer, 2022). The black edged circle and square are the mean of the 24-hourly values in water- and energy-limited**
**regimes, respectively. The colored numbers are the area within the curves (multiply displayed value by 10^6; units: [J/kg]^2); these**
**values quantify the diurnal energetic asymmetry captured by each mixing diagram. Dashed black lines are lines of constant relative**
**humidity. Note that x- and y-axis ranges differ. (b) The 3-hourly mixing diagrams in both climate regimes, computed with three**
**different starting times: hour 0 (S0: solid), hour 1 (S1: dashed dot), and hour 2 (S2: dashed) LST. (c) The 6-hourly mixing diagrams**
**as in (b), but with starting times at hours 0, 2, and 4 LST.**

## 4 Justifying our choices for how to reduce the data request

**Strategy for the reduction in time frequency**

To determine the optimal strategy for reducing the time frequency of the data request, yet still achieving the coupling
assessment goals discussed above, we consider two possible strategies: (i) regular, gridded time intervals, or (ii) time intervals
based on the local solar day (e.g., values for night-time, morning, and afternoon). Positive arguments for the first approach
include the lack of subjectivity and the ease of implementation. Counter-arguments centre around geographic differences



imposed by the gridded approach. For instance, in the summertime, sunrise times along one longitudinal band differ by about two hours between high-latitude regions and the equator. Thus, a 5 am data point on the summer equinox would be after sunrise in Sweden but before sunrise in the Congo, even though both have a longitude of 18°E. This poses difficulties for investigations of, for example, the Triggering Feedback Strength (TFS), meant to capture the impact of early-morning evaporative fraction on afternoon precipitation (Findell et al., 2011). Capturing specific times of day becomes more complicated with reduced frequency of data collection or archival. Since hourly data represent 15° longitudinal bands around the globe, coarser frequency data inherently require grouping broad longitudinal slices into common time points. For 6-hourly data, an attempt to capture early-morning conditions within one 90° longitudinal slice would produce local times that might span 6 time zones, potentially ranging from, for example, 3 am at the western edge to 8 am on the eastern edge. Clearly processes at the land surface and within the boundary layer differ substantially between these times of day. Higher-frequency data would reduce the severity of these issues, albeit with more archive space required.

Positive arguments for a data-archiving scheme linked to the solar day include fewer data points (and thus less archive capacity) needed to capture the three main phases of BL behavioural regimes (night-time, morning, afternoon), and a more uniform understanding of the solar conditions associated with each data point. However, any sub-daily selection based on the solar day requires *a priori* decisions that might be appropriate for one purpose, but which would restrict appropriateness for further study. For example, mixing diagrams are useful tools to understand BL evolution *within* each of the three solar day phases mentioned above. Saving average values within these three phases would eliminate the possibility of any sort of mixing diagram analysis of model behaviour. Additionally, interpretation of solar day-based data would be complicated by each archived data point representing different numbers of hours, both from day-to-day at one location, and from location-to-location on each day. Furthermore, this strategy would require additional code being written and implemented at each climate modelling centre, and thus the possibility of differences in implementation quickly emerges.

Here, we opt to make a request of regular, temporally gridded data to avoid the complications of solar day-based archiving and to maintain flexibility for future data usage. The negative features of the regular, gridded temporal data requests can be reduced with increased frequency of data storage. We propose 3-hourly data as a minimum request, with hourly or 2-hourly as improvements on that minimum. If this data request is still too cumbersome, a mask of oceanic regions can potentially be used to reduce the data volume by up to 2/3, though these data may be useful for the study of ocean-atmospheric boundary layer coupling processes.

**Other issues to confront**

In addition to decisions related to the reduction of the time frequency of data archiving, our data request must tackle difficult decisions related to (i) capturing mean BL properties while the height of the BL is changing, (ii) capturing adequate measures of the temperature and humidity gradients above the BL, and (iii) capturing soil conditions (moisture and temperature) most relevant to the partitioning of energy into surface fluxes of latent and sensible heat.





Determining average properties within the BL at any given time requires knowledge of the height of the BL ($h_{PBL}$). Model-computed $h_{PBL}$ is determined using different methods in different models, producing values which are self-consistent within each model's framework and, therefore, should adequately capture the time evolution of the BL height at a given location, and

the relative BL heights at different locations. However, night-time values of BL average properties will necessarily represent something different than daytime values, and values during the transition times of day will be tricky to compute and difficult to rely on. In addition, these times of day will change throughout the year. All of these issues suggest that care is needed in implementing these calculations and interpreting the results.

Finally, for characterization of soil conditions most relevant to surface energy partitioning, a root-zone soil moisture would be

most appropriate. However, since the root zone is both dynamic and dependent on vegetation type, no single depth can adequately capture the true root zone. Here we opt for a near-surface measure of the top 10 cm plus a slightly deeper measure averaged over the 10-100 cm interval. In both cases, we recognize that these are characterizations of the model's soil wetness, but that this variable is a model-specific quantity, different from in-situ or remotely sensed measures of soil wetness, and which should be interpreted with recognition of the model value's mean and variability (e.g., Koster et al., 2009, Benson and

Dirmeyer, 2023).

## 5 The Data Request

Here we present a concrete data request, dividing the request into three categories based on the analyses that would be enabled and by the work required by model developers. Request A is the highest priority request, and focuses on standard model output of surface fields saved at higher frequency intervals than is currently routinely practiced, thus requiring no additional work by

model developers, just additional archive space. This Request includes both Tier 1 and Tier 2 variables. Request B (second-tier priority) focuses on archival of variables in the lowest 300 mb of the troposphere. Like Request A, Request B requires no additional work by model developers, just additional archive space, while Request C (third-tier priority) requires in-code modifications to calculate average properties within and above the BL. After each request, we briefly mention which metrics (mostly from Figure 1) and analyses would become possible with these additional data.

The data length requirements of Findell et al. (2015) suggest that a minimum of ten years of data should provide for robust statistical analyses. Thus, for any simulation and/or time period of climatological interest, we request that these data are saved for at least a 10-year block of time. For historical and future scenario runs, it would be advantageous to have ten-year blocks saved at the beginning and end of the simulations.

**Request A: High-frequency archival of surface variables already included in standard model output:**
Table 1 details the variables included in Request A. The ten Tier 1 variables would allow for the computation of several two-legged metrics at sub-daily time scales (including all of those included in Figure 2), soil moisture memory, TFS and AFS,




basic mixing diagrams, and the percentile soil moisture—aridity index framework of Duan et al. (2023). Assuming a 1-degree grid without data compression, archival of Tier 1 variables would require approximately 13 GB/yr.

Request A also includes several Tier 2 priority variables: deeper soil moisture information, and component terms of net radiation and evapotranspiration. These additional terms would allow for more in-depth understanding model depictions of radiative processes and of the role of vegetation in driving evaporative fluxes and feedbacks. However, they would nearly double the required archival requirements, and, thus, have been deemed Tier 2 priority variables.

Of the ten Tier 1 variables listed in Table 1, the first eight were included at 3-hourly frequency in the HighResMIP data protocol

(Haarsma et al., 2016), with soil temperature (tsl) saved at 6-hourly frequency and boundary layer depth (bldep) saved monthly. HighResMIP also included 16 other variables in their 3-hourly data request (for a total of 24 3-hourly variables), indicating that saving all of the Request A variables is not an insurmountable challenge.

| Priority | Variable long name | Units | CMOR name | Frequency |
|---|---|---|---|---|
| 1 | Precipitation | kg m-2 s-1 | pr | 3hr |
| 1 | Surface upward sensible heat flux | W m-2 | hfss | 3hr |
| 1 | Surface upward latent heat flux | W m-2 | hfls | 3hr |
| 1 | Surface net radiation | W m-2 | * | 3hr |
| 1 | Near-surface (2m) air temperature | K | tas | 3hrPt |
| 1 | Near-surface (2m) specific humidity | 1 | huss | 3hrPt |
| 1 | Surface air pressure | Pa | ps | 3hrPt |
| 1 | Moisture in upper 10 cm of soil column | kg m-2 | mrsos | 3hrPt |
| 1 | Temperature of soil (in single near-surface layer) | K | tsl | 3hrPt |
| 1 | Boundary layer depth | m | bldep | 3hrPt |
| 2 | Components of surface net radiation:<br>Surface downwelling longwave radiation<br>Surface upwelling longwave radiation<br>Surface downwelling shortwave radiation<br>Surface upwelling shortwave radiation<br>Ground heat flux | W m-2 | <br>rlds<br>rlus<br>rsds<br>rsus<br>hfdsl | 3hr |
| 2 | Components of evapotranspiration:<br>Evaporation from canopy<br>Water evaporation from soil<br>Transpiration | kg m-2 s-1 | <br>evspsblveg<br>evspsblsoi<br>tran | <br>3hr<br>3hr<br>3hr |
| 2 | Moisture in 10-100 cm of soil column | kg m-2 | * | 3hr |



**Table 1: Specifics of Request A. Grid cell average values are either 3-hourly time means (3hr) or at an instantaneous point in time at the end of the time interval (3hrPt). The * symbol indicates variables without standard Climate Model Output Rewriter (CMOR) names.**

**Request B: High-frequency archival of data at several specified lower-tropospheric pressure levels:**

Table 2 details the five variables included in Request B for archival of select lower tropospheric pressure levels, specifically

temperature, humidity, and three-dimensional winds. The priority here is to enable systematic exploration of BL processes throughout various stages of growth, development, and decay. Saving high-frequency data of full atmospheric profiles is not realistic, but saving a few select pressure levels would allow for the computation of atmospheric stability and humidity deficit in the early-morning hours (i.e., metrics like CTP and $HI_{low}$), mean properties within the BL, $d\theta/dz$ and $dq/dz$ above the BL, the heated condensation framework, and more complex mixing diagrams than Request A would enable, including identification

of advection and entrainment terms during multiple phases of BL growth and development. The six specific pressure levels requested are every 50 hPa between 950 and 700 hPa.

| Priority | Variable long name | Units | CMOR name | Frequency |
|---|---|---|---|---|
| 2 | Eastward Wind at six pressure levels | m s-1 | ua | 3hrPt |
| 2 | Northward Wind  at six pressure levels | m s-1 | va | 3hrPt |
| 2 | Omega (=dp/dt) at six pressure levels | Pa s-1 | wap | 3hrPt |
| 2 | Air Temperature at six pressure levels | K | ta | 3hrPt |
| 2 | Specific humidity at six pressure levels | 1 | huss | 3hrPt |

**Table 2: Specifics of Request B. The six requested pressure levels are every 50 hPa between 950 and 700 hPa. Grid cell average values are instantaneous in time at the end of the time interval (3hrPt).**


**Request C: Variables requiring code modifications for internal computation:**

With Request C, we aim to enable more accurate mixing diagram work than is possible with Request B, while simultaneously reducing the archive requirements needed to assess mean properties within and above the BL. Request C entails code modifications to determine, at each time step, the BL mean thermal and moist energy content per unit mass ($c_p\theta$ and $\lambda q$,

respectively), changes of these terms due to advection, and the mean potential temperature and humidity gradients across the top of the BL, given by $h_{PBL}$ (or the CMOR variable name *bldep* in Table 1). For a standard definition of $h_{PBL}$, we suggest the Bulk Richardson number definition of Seidel et al. (2012), consistent with the data available in reanalyses such as ERA5 and MERRA2. Specifically, we request the mean BL properties vertically integrated from $0.1*h_{PBL}$ to $0.8*h_{PBL}$, and mean $\theta$ and $q$ gradients over the interval closest to $0.8*h_{PBL}$ to $1.2*h_{PBL}$, given model level constraints (see Turner et al., 2014 for selection

of these vertical bounds). These properties should be saved every three hours.



While Request C would reduce the archive requirements for mixing diagram work and provide a fuller picture of mean mixed layer behaviour, it would not allow for some of the other metric calculations that Request B does cover. Thus, these are complementary requests, rather than substitutes for each other.

| Priority | Variable long name | Units | Frequency |
|---|---|---|---|
| 3 | Mean BL heat content | J kg-1 | 3hrPt |
| 3 | Mean BL moisture content | J kg-1 | 3hrPt |
| 3 | BL heat advection tendency | W kg-1 | 3hr |
| 3 | BL moisture advection tendency | W kg-1 | 3hr |
| 3 | BL-top temperature gradient | J kg-1 | 3hrPt |
| 3 | BL-top moisture gradient | J kg-1 | 3hrPt |

**Table 3: Specifics of Request C. The BL-mean properties should be vertically integrated from 0.1\*hPBL to 0.8\*hPBL, while the gradients across the BL top should be calculated over the interval 0.8\*hPBL to 1.2\*hPBL. CMOR names are not currently available for these quantities.**

**6 Conclusions**

Increasing the time resolution of model output describing components of land-atmosphere coupling and processes within the
land-atmosphere interface is essential to fully and accurately model, understand, and predict these processes, and to compare modelled processes with observational datasets. The data request described here will allow us to compare coupled earth system and climate models with observations from field campaigns and compare both diurnal and long-term properties of L-A interactions in different models and during model development. These sorts of comparisons are essential to fully assess the land-atmosphere coupling behaviours of different GCMs. Furthermore, these improvements to our understanding of processes
at the land surface are essential to understanding the vulnerability of humans and ecosystems to changing climatic conditions and improving our resiliency in the face of a likely increase in extremes.

**Code and data availability**

The Copernicus Climate Change Service (C3S) provides access to ERA5 data freely through its online portal at https://cds.climate.copernicus.eu/cdsapp#!/dataset/reanalysis-era5-single-levels (Hersbach et al., 2020).
The code for calculating two-legged metrics, TFS, and mixing diagrams can be found at https://github.com/abtawfik/coupling-metrics (Tawfik, 2023).
The source code for calculating diurnal mixing diagram is shared on GitHub (https://github.com/ekseo/CLASP_LoCo.git, last access: 06 July 2023; https://doi.org/10.5281/zenodo.8117559, ekseo, 2023).



The source code for data analysis and visualization of Figure 3 and 4 as well as the corresponding diagnostic results (i.e.,

triggering feedback strength and two-legged metrics based on ERA5 reanalysis data) are freely available on GitHub (https://github.com/yinzun2000/CLASP_LoCo, last access: 21 August 2023; http://doi.org/10.5281/zenodo.8304156).

Flux tower observations used for Figures 2 and 5 are openly available from the FLUXNET2015 Tier 1 data (https://fluxnet.org/data/download-data/, Pastorello et al., 2020), the AmeriFlux network (https://ameriflux.lbl.gov/data/download-data/, Novick et al., 2018), the drought-2018 network

(https://doi.org/10.18160/YVR0-4898, and Drought 2018 Team and ICOS Ecosystem Thematic Centre, 2020).

**Competing Interests**

Po-Lun Ma and David Lawrence are Topical Editors of Geoscientific Model Development. Other authors declare that they have no conflict of interest.

**Author Contributions**

The manuscript was originally conceived during meetings of the diagnostics team of the Coupling Land and Atmospheric Subgrid Parameterizations (CLASP) project. Input was sought from contributors to other aspects of the CLASP project. KLF, ZY, ES, and PAD contributed figures. KLF prepared the manuscript with contributions from all authors.

**Acknowledgements**

This study was supported by NOAA's Climate Program Office's Modeling, Analysis, Predictions, and Projections program

and the Department of Energy, Office of Science, Biological and Environmental Research program, Earth System Model Development program area, as part of the Climate Process Team (CPT) on Coupling Land and Atmospheric Subgrid Parameterizations (CLASP). This included support of ZY at Princeton's Cooperative Institute for Modeling the Earth System (CIMES), NOAA grant NA18OAR4320123; PAD at the Center for Ocean-Land-Atmosphere Studies at George Mason University, NOAA grant NA19OAR4310242; NC at Duke University, NOAA grant NA19OAR4310241; MF at NCAR,

NOAA grant NA19OAR4310241 and DOE project #4000178550; and PLM and MH at the Pacific Northwest National Laboratory (PNNL), DOE project no. 73742. PNNL is operated for the DOE by the Battelle Memorial Institute under Contract DE-AC05-76RL01830. DML is supported by the National Center for Atmospheric Research (NCAR), which is a major facility sponsored by the NSF under Cooperative Agreement 1852977. JAS contributions at NASA's Goddard Space Flight Center was supported by David Considine (NASA HQ) under the NOAA CPT. ES was supported by the Korea Meteorological

Administration Research and Development program under grant RS-2023-00241809. ZY, PAD, and NC had additional funding through NOAA grants NA22OAR4050663D, NA22OAR4310643, and NA220AR0AR4310644 respectively.



We thank the European Centre for Medium-Range Weather Forecasts (ECMWF) for providing the ERA5 data.

We thank Randy Koster, Mitch Bushuk, and Wenhao Dong for providing valuable feedback on the manuscript. We thank Catherine Raphael for graphics help with Figure 1.

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
