# Peer review of "Accurate Assessment of Land-Atmosphere Coupling in Climate Models Requires High Frequency Data Output"

_EGUsphere, 2023_

## Author Response (AR1)

**Peer-review #1, from Divyansh Chug**

Accurate Assessment of Land-Atmosphere Coupling in Climate Models Requires High Frequency Data Output

by Kirsten Findell et al.

This study outlines a practical data request that would allow climate model developers, users and educators to adequately characterize (and diagnose the shortcomings of) the sub-daily coupling processes between the land and the atmosphere, in their numerical model of choice. Typically, climate model outputs have enabled such characterization through monthly mean (or in some case, daily mean) data which is inadequate the capture land-atmosphere (L-A) interaction processes, specifically related to daytime boundary layer development. The clear outline provided in this paper on the specific variables, temporal resolution, and length of dataset required for L-A coupling diagnosis, using the Local L-A Coupling (LoCo) framework, offers a consistent guideline for the research community. The authors have provided multiple use-cases that illustrate the utility of their request. It's clear that they have carefully optimized the request with regards to the marginal storage space and effort needed to perform this additional task.

The claims made by the authors in this research article are as follows:

1.  No single metric currently in practice captures all the modes, means, and methods of interaction between the land and the atmosphere.
2.  The typical resolution of Earth System Model output (daily; or 6-hourly at best) is insufficient for characterizing model behavior for important sub-daily processes captured by the LoCo metrics.
3.  Higher-frequency model output is needed to ensure model fidelity, robustness and further development.

This paper builds on the previous literature (with some additional and modified concepts) summarized by Santanello et al. (2018). It provides helpful considerations on how to apply and interpret the coupling metrics based on the temporal resolution of the dataset. Unlike previous efforts, this work provides a clear outline for the ingredients required to effectively perform this task (of characterizing L-A interactions). This is a significant stride toward standardizing the analysis and diagnosis of model behavior relevant for the L-A interactions research community, specifically for those whose research can benefit from the LoCo metrics. I found zero inconsistencies or flaws in the manuscript. As such, this manuscript merits publication as is.

Citation: https://doi.org/10.5194/egusphere-2023-2048-RC1

*Reply: We are happy to read that the reviewer understood and appreciated the goals of this manuscript. We do hope that it inspires enhanced assessment and evaluation of land-atmosphere coupling behavior in Earth system models.*

**Peer-review #2, from Timothy Lahmers:**

This data-request provides a valuable addition to the discipline by outlining existing PBL knowledge and discussing current limitations of existing datasets, given their temporal resolution and missing values. This research is especially relevant in a changing climate, and the authors note that this work is relevant to the understanding of hot/dry extremes, as well as wet extremes.

The authors do a good job outlining the need for higher temporal resolution, considering this in terms of physical processes and variability through the diurnal cycle, and they consider this using the Mixing Diagram framework, to show the limitations of coarser data.

While this manuscript will be an important contribution, I have some technical and structural concerns for the authors:

- While the authors are careful to address the precise needs for different levels of temporal resolution in their data-request, there is little information about spatial resolution. Since PBL processes occur on the scale of meters to the meso-beta scale, this request would be stronger with more details on the spatial scale of the data required.
- Related to this above point, the authors note that a request for 1-degree spatial resolution data would require 13 GB per year (lines 350 to 354). Is 1-degree spatial resolution appropriate, given that it is now coarser than most global models and would likely be unable to capture most mesoscale processes (e.g., individual thunderstorms and surface gradients across fronts)? Would this resolution be sufficient to resolve PBL process or is a higher resolution required?
- Figure 2 is a useful conceptual illustration for the reader to evaluate PBL linkages for model simulations compared to observations; however, the blue dashed lines are difficult to see. Could the authors update this figure to make this component more legible?

Citation: https://doi.org/10.5194/egusphere-2023-2048-RC2

*Reply: We are happy to read that this reviewer also understood and appreciated the goals of this manuscript.*

*As the review points out, the focus here is on enhanced temporal data output. As mentioned in the abstract, this paper grew out of a project focused on parameterizing and characterizing the impact of subgrid heterogeneity on ESM behavior and performance, so we fully appreciate the importance of the reviewer's first two concerns. We note in the abstract that this manuscript enables a necessary step towards achieving that aim, but we agree with the reviewer that the manuscript will be strengthened if we broaden our discussion of spatial scales.*

*In response to the first request, we have added the following text:*

> The spatial scales of individual grid cells in ESM simulations included in the most recent Coupled Model Intercomparison Project (CMIP6) typically range from 50 km to 250 km, with models run at resolutions finer than 50 km eligible for participation in the High-Resolution Model Intercomparison Project (HighResMIP; Haarsma et al., 2016). These

resolutions suggest that the footprint sampled from in situ observations (ranging from cm-scale soil moisture probes to wind- and height-dependent flux tower sampling fetches on the order of hundreds of meters) is substantially smaller than individual ESM grid cells. This suggests that, when possible, observational comparisons should be made against sub-grid tiles representing fractional areas of differing land use types. However, saving tile-specific high-frequency data is likely not feasible for most modelling centres. Given that reality, the data request outlined here will enable previously impossible assessment of grid-cell mean behaviour throughout the diurnal cycle. Future work motivated by the CLASP project can extend these lines of inquiry to issues centred on sub-grid spatial heterogeneity, or to comparisons with global storm-resolving efforts like those of Stevens et al. (2019).

*In response to the second point, archive requirements for 1-degree information were provided just to give readers a sense of the storage space needed to meet this data request. To make this clearer, we added the phase "(for reference)" after "Assuming a 1-degree grid."*

*In response to the third point, we have reformatted Figure 2 so the panels are arranged vertically, allowing them to be bigger so the lines (even the slimmest ones with the weakest correlations) are more visible. This also addresses a point made by Reviewer #3.*

**Peer-review #3, Anonymous:**

**Main comments:**

1. Implications for climate variability and change not discussed - specifically with reference to Figure 2 (b) and (c) authors show improvement in UFS P6 to UFS P7; I am curious how does these improvements reflect not reduction in mean biases and water cycle predictability. Authors may consider showing the lead-lag correlation between soil moisture and precipitation between P6 and P7 model. If authors find a positive results, then it can be a better sell to CMIP groups.

2. There is lot of self-citation; like glace metric, loco, mixing diagram etc. In the process, authors missed many new literature that show longer soil moisture memory effects and its effect on water cycle predictability, e.g. https://www.nature.com/articles/s41612-021-00172-z. Hence soil moisture memory oval in figure 1 should extend to months and inter-annual time scale too!

**Minor comment:**

Figure 2 is too small to read anything legibly; so authors may consider dividing it into two parts.

Citation: https://doi.org/10.5194/egusphere-2023-2048-EC1

*We thank this reviewer for these helpful suggestions.*

*Main comment #1: We agree that evidence of improved lead-lag correlations between soil moisture and precipitation in moving from UFS P6 to UFS P7 would be a good selling point. We feel, however, that this requires a full assessment that is beyond the scope of this paper. In fact, three of our co-authors are involved in a separate paper does just that:*

- *Eunkyo Seo, Paul A. Dirmeyer, Michael Barlage, Heiln Wei, and Michael Ek, 2023: Evaluation of land-atmosphere coupling processes and climatological bias in the UFS global coupled model. Journal of Hydrometeorology, DOI: https://doi.org/10.1175/JHM-D-23-0097.1.*

*In that manuscript, the authors investigate the performance of the "NCEP Unified Forecast System (UFS) Coupled Model prototype simulations (P5–P8) during boreal summer 2011–2017 in regard to coupled land-atmosphere processes and their effect on model bias."*

*Main comment #2: Thank you for this additional reference. This is a very nice paper with promising implications for possible improvements to drought prediction systems. We have modified Figure 1 to extend the soil moisture memory oval out to the 'Years' marker on the x-axis. We have also added the following text to the caption for Figure 1:*

> "Esit et al. (2021) show promising predictability benefits from soil moisture initialization, extending the scope of soil moisture memory into the seasonal-to-decadal time frame."

*In response to the minor comment, we have reformatted Figure 2 so the panels are arranged vertically, allowing them to be bigger so the lines (even the slimmest ones with the weakest correlations) are more visible. This also addresses a point made by Reviewer #2.*